# Characteristics and Relationships between Total Polyphenol and Flavonoid Contents, Antioxidant Capacities, and the Content of Caffeine, Gallic Acid, and Major Catechins in Wild/Ancient and Cultivated Teas in Vietnam

**DOI:** 10.3390/molecules28083470

**Published:** 2023-04-14

**Authors:** Tran-Thi Nhu-Trang, Quoc-Duy Nguyen, Nguyen Cong-Hau, Le-Thi Anh-Dao, Philippe Behra

**Affiliations:** 1Faculty of Environmental and Food Engineering, Nguyen Tat Thanh University, Ho Chi Minh City 700000, Vietnam; nqduy@ntt.edu.vn (Q.-D.N.);; 2Laboratoire de Chimie Agro-Industrielle (LCA), Université de Toulouse, INRAE, Toulouse 31400, France; philippe.behra@ensiacet.fr

**Keywords:** tea, total polyphenol content, total flavonoid content, antioxidant activity, caffeine, gallic acid, catechins, correlation coefficient, principal component analysis

## Abstract

Vietnam has diverse and long-established tea plantations but scientific data on the characteristics of Vietnamese teas are still limited. Chemical and biological properties including total polyphenol and flavonoid contents (TPCs and TFCs), antioxidant activities (DPPH, ABTS, FRAP, and CUPRAC), as well as the contents of caffeine, gallic acid, and major catechins, were evaluated for 28 Vietnamese teas from North and South Vietnam. Higher values of TPCs and TFCs were found for green (non-oxidised) and raw Pu’erh (low-oxidised) teas from wild/ancient tea trees in North Vietnam and green teas from cultivated trees in South Vietnam, as compared to oolong teas (partly oxidised) from South Vietnam and black teas (fully oxidised) from North Vietnam. The caffeine, gallic acid, and major catechin contents depended on the processing, geographical origin, and the tea variety. Several good Pearson’s correlations were found (r^2^ > 0.9) between TPCs, TFCs, the four antioxidant capacities, and the content of major catechins such as (–)-epicatechin-3-gallate and (–)-epigallocatechin-3-gallate. Results from principal component analysis showed good discriminations with cumulative variances of the first two principal components varying from 85.3% to 93.7% among non-/low-oxidised and partly/fully oxidised teas, and with respect to the tea origin.

## 1. Introduction

Measuring the antioxidant activities of food products is becoming increasingly important worldwide, since such measurements can provide a crucial information on the antioxidant capacities as well as the impact of this antioxidant activity on organisms that absorb these products [1,2]. Among food products, tea (*Camellia sinensis*), being one of the most popular beverages consumed, has spurred many studies on its chemical aspects and biological activities, mainly due to the increased interest in its health benefits along with its popularity [3,4,5].

Based on their mechanism of action, antioxidants are classified into two main groups: primary and secondary/preventive antioxidants. Primary antioxidants directly disrupt the free radical scavenging chain reaction by transferring electrons or hydrogen atoms from hydroxyl groups. This stabilizes the free radicals, inhibits/retards the initiation phase, and terminates the propagation phase of autoxidation [6]. Secondary antioxidants can chelate with metal ions (such as iron and copper), absorb ultraviolet radiation, capture oxygen, and regenerate primary antioxidants. Antioxidants can act through a variety of mechanisms; hence, it is important to measure the antioxidant capacity using different methods [2,7,8]. Free radical scavenging methods using 2,2-diphenyl-1-picrylhydrazyl (DPPH) and 2,2′-azino-bis(3-ethylbenzothiazoline-6-sulfonic acid) (ABTS) have been developed to determine the ability of the antioxidant to transfer hydrogen to free radicals. Similarly, the metal ion reducing capacity determines the ability of the antioxidant to transfer electrons to ferric and cupric ions [9].

In general, research on tea has been very diverse, focusing not only on the content and activity of the active ingredients in different teas and their infusions, but also on analysing the composition changes and the formation of new compounds during processing of the tea sample. Several studies have been conducted to determine the antioxidant activity of tea based on multiple antioxidant mechanisms. However, most studies have only identified one or two indicators to illustrate the antioxidant potential of tea [10]. The content of polyphenol compounds and the antioxidant activities of three different teas, namely green, black, and yellow teas, have been compared using different analytical methods [11]. In this work, the authors comprehensively evaluated the quality of teas with respect to the different characteristics, including total polyphenol composition and the seven tea-specific phenolic compounds, antioxidant activity (DPPH, FRAP, CUPRAC), ferrous chelation capacity, and the sensory quality of the teas and their infusions. The total polyphenol contents in green tea, black tea, and yellow tea were reported to be 525.87, 668.49, and 374.11 mg GAE L^–1^, respectively. Among these teas, the FRAP activity, %DPPH inhibition, and CUPRAC activity were the highest for black tea with values of 25.02, 56.40, and 11.42 (mmol TE L^–1^), respectively. To date, there is no officially recognised standardised method to evaluate the antioxidant activity of food products [12] and it is, therefore, very difficult to compare and compile the antioxidant activity of tea from different studies, since the tea varieties, extraction methods, and evaluation protocols adopted in these studies have been very different. Therefore, we believe that it is important to use various analytical methods to determine the antioxidant capacity so as to help researchers in this field to obtain more information on the overall antioxidant capacity of the tea sample in question. 

Apart from the cell wall compounds, polyphenols, being the most important ingredient in tea, have been considered as important chemical compounds that account for the health benefits of tea [13]. The predominant polyphenols in tea leaves are flavonoids belonging to the flavan-3-ol group, (typically catechins, which comprise one-third of the tea dry mass) along with other flavonoid components such as quercetin, myricetin, and kaempferol [14,15]. Four catechins including (–) epigallocatechin-3- gallate (EGCG), (–) epicatechin-3-gallate (ECG), (–) epigallocatechin (EGC), and epicatechin (EC) account for 30–50% of the composition in green tea extracts and nearly 10% in black tea extracts, with black tea consisting predominantly of polymerised catechins (theaflavins and thearubigins) [15]. The catechin content in green tea makes up 85% of the total polyphenol after harvest [16]. In the processed teas, the polyphenol content, which has a notable impact on their antioxidant power, depends on many factors such as the tea variety and the processing method used [17,18,19]. The content and activity of phenolic compounds, which are among the important indicators of tea quality, are reported to be influenced by many environmental factors (air temperature, rainfall, light intensity) and cultivation variables (fertilisation, soil, plucking frequency). Other factors such as geographic location and harvest season have also been reported to play a role [20,21,22]. 

Vietnam has a diverse and long-established tradition of tea cultivation with a wide variety of teas. Tea trade has been ongoing for hundreds of years and currently, Vietnam produces one million tonnes of tea per year for both local consumption and export [23]. However, there are insufficient scientific data on the characteristics of Vietnamese teas and Vietnam remains a largely unstudied tea region, as compared to other tea cultivation and exploitation areas in the world such as China, India, and Japan. Thus, the three main objectives of this study are (i) to report data on total phenolics, flavonoids, the four antioxidant activities (DPPH, ABTS, FRAP, and CUPRAC), the contents of caffeine, gallic acid, and major catechins present in green, raw Pu’erh, oolong, and black teas of different geographical origins in North and South Vietnam including both wild/ancient and cultivated tea plants; (ii) to elucidate, based on chemical and biological properties, the characteristics and the relationships among the studied parameters to better understand the impact of tea processing conditions and the variety of tea on the composition of the dried tea products; and (iii) to differentiate the different Vietnamese teas with respect to their geographical origin, variety, tree age, and processing conditions by principle component analysis.

## 2. Results and Discussion

### 2.1. TPCs and TFCs

Among the 28 dried tea products that were collected in 2020 and 2022 (Appendix A), 11 were chosen as representative samples to analyse TPCs, TFCs, and the 4 antioxidant capacities (Table 1). Tea samples collected from Suoi Giang commune (Yen Bai Province, North Vietnam) included three types of tea: green, raw Pu’erh, and black teas. They were produced from wild/ancient tea trees that are more than 200 years old. The results presented in Table 1 show that green and black teas from this region had the highest and lowest TPCs of 287.2 mg GAE g^–1^ DM for SG-G-01 and 151.4 mg GAE g^–1^ DM for SG-B-01, respectively, while the highest and lowest TFCs are 207.1 mg QE g^–1^ DM for SG-G-07 and 128.8 mg QE g^–1^ DM for SG-B-01, respectively. The TPCs and TFCs of green and raw Pu’erh teas are rather comparable. The decline in TPCs and TFCs has been explained to be due to the difference in degree of oxidation during processing among the tea samples; the green tea samples were referred to as non-oxidised tea due to their deactivation of enzymes at high temperature [19,24,25]. Raw Pu’erh teas were lightly oxidised but would be fermented during ripe Pu’erh processing [26,27] and black teas were totally oxidised.

Among the tea samples in Da Lat city and Lam Ha district (Lam Dong Province, South Vietnam), there were two types of teas originating from two varieties of tea trees and subjected to two different processing conditions to obtain for green and oolong teas. The variation in TPCs and TFCs of these teas was similar to that in Suoi Giang, in which the green tea (non-oxidised) named DL-G-01 showed 217.2 mg GAE g^–1^ DM for TPCs and 125.3 mg QE g^–1^ DM for TFCs, while oolong teas (partly oxidised) named DL-O and LH-O showed 165.4 to 179.2 mg GAE g^–1^ DM for TPCs, and 67.7 to 73.7 mg QE g^–1^ DM for TFCs (Table 2). The observed variations in TPCs and TFCs are consistent with results published in previous studies [31,32]. 

In general, raw Pu’erh and green tea samples in our study had higher TPCs and TFCs than black and oolong teas. Green teas in SG and DL underwent mostly similar processing but their tea trees were different with respect to variety/cultivar (var. Shan and var. Assam, respectively) and age (>200 years old for SG tea trees and around 100 years old for DL). These differences led to a difference in TPCs and TFCs for these green teas, with higher values obtained for wild/ancient teas in Suoi Giang than cultivated ones in Da Lat. In the case of oxidised teas, oolong and black teas have higher levels of oxidation (partial and full, respectively) leading to lower TPCs and TFCs. During oxidation, the enzyme polyphenol oxidase oxidises simple catechins to form low-molecular-mass dimers (theaflavins) or high-molecular-mass polymers (thearubigins) leading to a reduction in TPCs and TFCs [28,33]. The oxidation of catechins to theaflavin compounds during oxidation/fermentation also contributes to the intense colour and characteristic flavour of oxidised tea [34]. A comparison of our studied teas with some tea samples worldwide is presented in Table 1, TPCs of Vietnamese tea samples in this current study, regardless of the tea type, were significantly higher than some teas originating from Asian countries such as Malaysia and China [28,30]. However, the TPCs of Vietnam’s green and black teas were comparable to those of the same tea type from Kenya, while Vietnamese oolong teas had lower TPCs [29].

The TFCs of green teas and raw Pu’erh teas in Suoi Giang (Yen Bai Province, North Vietnam) were rather similar (183.6 and 188.0 mg QE g^–1^ DM for SG-P samples and 177.3 to 207.1 mg QE g^–1^ DM for SG-G). These two types of teas that were non-oxidised and low-oxidised, respectively, still retained a high concentration of catechins. Black tea samples from Suoi Giang (North Vietnam) had higher TFCs than oolong teas from Da Lat and Lam Ha (South Vietnam), (73.7 mg QE g^–1^ DM and 67.7–70.9 mg QE g^–1^ DM, respectively), although the oxidation level of oolong teas was lower than black teas. This difference in TFCs between the black and oolong tea samples can be explained to be due to the difference between the tea varieties, cultivars (organic or conventional agriculture), and growing conditions (soil composition, altitude, climate as well as processing conditions [15,32,35,36]. These results prove that the tea samples from wild/ancient tea trees in Suoi Giang possess specific properties in terms of their organic composition.

The difference in TPCs and TFCs between oolong and green teas in Lam Dong province (South Vietnam) was quite significant, with lower values measured for oolong teas. This difference can be attributed to the different processing methods; the oolong teas tested here went through many stages of drying and rolling as well as a longer processing time with a total processing time of 36 to 48 h. Traditional drying processes, such as air oven drying, especially at high temperatures and long drying times, could destroy the heat-labile polyphenols, thus reducing the nutritional value, taste, texture, and antioxidant activity of the tea [37]. The loss of phenolics or flavonoids during high-temperature drying can be attributed to the release of bound polyphenols as well as the release of phenolic acid from lignin, followed by their thermal decomposition [37,38]. In addition, for the green tea, the high drying rate (short heating time) also disactivated the oxidative enzymes present in the plant material, as a result of which, the polyphenol compounds were better protected [39]. A 23% [40] and 30% [41] decrease in flavanols has also been observed in black tea, where kaempferol and quercetin glycosides did not change significantly compared to fresh leaves, whereas the myricetin triglycosides completely disappeared and a 50% decrease is observed for the monoglycoside [40].

### 2.2. Antioxidant Capacities

#### 2.2.1. DPPH and ABTS

For the tea samples from Suoi Giang commune (Yen Bai Province, North Vietnam), the DPPH free radical scavenging activity decreased in the order of green teas (2754.0 to 2787.4 mol TE g^–1^ DM)~raw Pu’erh teas (2482.0 and 2513.5 µmol TE g^–1^ DM) followed by black teas (1745.4 and 1803.8 µmol TE g^–1^ DM) (Figure 1). Additionally, similar to TPCs and TFCs, green teas and raw Pu’erh teas showed only a minor difference in DPPH, presumably due to the similar tea variety and non or low oxidation during the processing of these two types of tea. For the teas from Da Lat city (Lam Dong Province, South Vietnam), the green tea sample showed higher DPPH value (2541.3 µmol TE g^–1^ DM) than the oolong one (2074.0 µmol TE g^–1^ DM). These results prove that both the extent of oxidation during processing and the tea variety affect the DPPH free radical scavenging activity.

However, when compared with TFCs, the decrease in DPPH was lower for oxidised and non-oxidised teas. This result shows that the DPPH free radical scavenging antioxidant activity is strongly related to certain specific compounds responsible for this activity in teas. Concerning oolong teas grown in Lam Ha district and Da Lat city belonging to Lam Dong Province (South Vietnam), the tea trees are similar in variety and age, but are grown under dissimilar cultivation conditions (organic agriculture for LH and conventional cultivation for DL), altitude (1200 m and 1600 m for LH and DL, respectively), and climate. These factors could account for the slight difference in DPPH between oolong teas in Lam Ha and Da Lat, which were 1809.8–1835.6 mol TE g^–1^ DM and 2074.0 mol TE g^–1^ DM, respectively. When comparing green teas in Da Lat and Suoi Giang, the variation in DPPH values was negligible even though there were differences in tea varieties, cultivation conditions (conventional agriculture vs. wild-growing), soil compositions, climate, and latitude (2541.3 µmol TE g^–1^ DM for DL-G-01 and 2754.0 to 2787.4 µmol TE g^–1^ DM for SG-G). Several previous studies have shown that differences in weather conditions, climate, cultivation conditions, and processing methods influenced the properties and characteristics of the teas and this impact could be discerned and used to differentiate the tea types based on the DPPH free radical scavenging activity [25,42,43,44].

With the ABTS free radical scavenging mechanism, for the same tea variety in Suoi Giang commune, the antioxidant capacity decreased gradually in the order of green teas (from 4506.2 to 4675.1 µmol TE g^–1^ DM), raw Pu’erh teas (4431.7 and 4474.3 mol TE g^–1^ DM), and black tea (3283.6 and 3343.4 mol TE g^–1^ DM) (Figure 2). This variation is consistent with some previous studies in which green teas showed the highest antioxidant activity, followed by oolong tea and black tea [20,45]. For samples coming from Da Lat, ABTS values of green and oolong teas were almost similar (3651.2 µmol TE g^–1^ DM and 3505.6 µmol TE g^–1^ DM, respectively) even though these two were two different types of tea in terms of variety, age, and processing. For oolong teas from the two regions DL and LH, similar to the DPPH capacity, there was no significant difference in ABTS values (3025.7 to 3505.6 mol TE g^–1^ DM). However, there was a remarkable difference in ABTS values between wild/ancient green teas in Suoi Giang and cultivated ones in Da Lat.

These results show that the antioxidant capacities obtained by ABTS mechanism are higher than those by DPPH, likely because of the chemical structure of these two free radicals. Specifically, while DPPH is a nitrogen-based radical containing an unpaired electron, the ABTS radical acts as both electron and proton acceptor due to its cationic nature [46]. In addition, the ABTS data more clearly differentiated tea regions and tea types. An important difference between ABTS and DPPH reactivities is that the DPPH reagent does not react with flavonoids that do not contain a hydroxyl group on both the B ring and the aromatic ring, while ABTS reagent is less selective and reacts with any hydrogen donor. Notwithstanding, this difference is apparent for tea but not observed in other food matrices with high lipid contents and cereal-based products where DPPH values were higher than ABTS ones [47,48,49]. Therefore, it can be concluded that the antioxidant activity also depends on the reaction environment, the stability of free radicals, and the phenolic radical stability.

#### 2.2.2. CUPRAC and FRAP

The ability of antioxidants to reduce metal ions in the FRAP assay is subject to the electron transfer mechanism. Hence, when used along with other assays, this method is useful to elucidate the dominant mechanisms operative in the antioxidant response. In contrast, the CUPRAC method has the advantage in determining the antioxidant capacity in vivo, since the reaction takes place in a neutral pH medium, close to the physiological pH in organisms (7.4). Moreover, this method can also be used to evaluate the activity of thiol-type antioxidants, such as glutathione, in biological cells [50,51]. In addition, CUPRAC is more reactive than FRAP due to its low redox potential, close to the ABTS^+^/ABTS pair, which means that it does not react with non-phenolic reducing agents such as sugar or citric acid. Similar to the ABTS assay, CUPRAC is also capable of measuring the antioxidant capacity of hydrophilic and lipophilic substances [51,52,53].

The results presented in Table 1 show a higher antioxidant activity of tea samples following the CUPRAC mechanism as compared to those with the FRAP mechanism. In addition, tea samples belonging to the same tea region and the same cultivar with the same processing method exhibited a similar variation trend. This is consistent with previous studies [10,25]. For the ancient tea variety from Suoi Giang commune, FRAP and CUPRAC values decreased in the order of non-oxidised green teas (2334.4 to 2927.6 µmol TE g^–1^ DM and 4121.1 to 4410.6 µmol TE g^–1^ DM, respectively) > low-oxidised raw Pu’erh teas (2721.4–2806.3 µmol TE g^–1^ DM and 4060.7–4201.1 µmol TE g^–1^ DM, respectively) > fully oxidised black teas (1906.6–2043.4 µmol TE g^–1^ DM and 2927.3–2928.8 µmol TE g^–1^ DM, respectively). This variation shows that oxidation affected the FRAP and CUPRAC values; more specifically, the degree of oxidation likely reduced antioxidant activity when following the transition metal reduction mechanism [54,55]. Similar results have been reported in other studies [36,56]. During processing, oxidation alters the chemical composition of tea, especially with respect to compounds with antioxidant properties. Such alteration occurs through decomposition, dimerisation, and polymerisation to form complex compounds with larger molecular mass. The formation of these new compounds is expected to reduce reactivity in the transition metal reduction reaction.

Among the teas from Da Lat city (Lam Dong Province, South Vietnam), green tea showed FRAP and CUPRAC values of 2361.1 µmol TE g^–1^ DM and 3625.5 µmol TE g^–1^ DM, respectively, while the corresponding values for the oolong tea were 1561.1 µmol TE g^–1^ DM and 3079.2 µmol TE g^–1^ DM, respectively. For oolong teas from Lam Ha district (Lam Dong Province, South of Vietnam), the FRAP and CUPRAC values ranged from 1481.2 to 1552.0 µmol TE g^–1^ DM and from 2704.9 to 2726.2 µmol TE g^–1^ DM, respectively. No significant difference in values was observed between oolong teas from DL and LH. Thus, in addition to the effect of oxidation during processing, the difference in FRAP and CUPRAC values here could also be due to differences in tea cultivars (as described above), cultivation conditions, and growth conditions (altitude and climate).

### 2.3. Determination of Caffeine, Gallic Acid, and Major Catechins by UPLC-MS/MS

The contents of caffeine (CFI), gallic acid (GA), catechin (C), epicatechin (EC), epicatechin gallate (ECG), epigallocatechin (EGC), and epigallocatechin gallate (EGCG) in tea change depending on variations in processing conditions such as oxidation level, time, and temperature at each production stage, and mechanical impacts (cutting, bruising crushing, rolling, etc.). Several studies have been conducted for Chinese teas [18,19,57,58], which are considered to have characteristics close to Vietnamese teas. Although the tea production processes in Vietnam are mostly similar to those in China, there are some small differences in procedure in each production stage, mainly due to the taste preferences of consumers in the two countries. In addition, climatic conditions and tea varieties also contribute to the difference in composition.

Lin et al. [19] remarked that the level of caffeine in different manufactured teas was in the order black tea > oolong tea > green tea > fresh tea leaf. However, in the current study, the concentrations of caffeine varied from 13.6 to 54.4 mg g^−1^ and did not depend on the type of tea (Table 2). In general, old/wild teas had higher caffeine contents. The formation of gallic acid depends on the withering process and oxidation, and its concentrations in the 28 samples were very different and were not specific to each tea. Regarding the major catechins, the EGCG concentrations were quite similar (41.5 to 67.5 mg g^−1^) for oolong teas of Da Lat and Lam Ha (South Vietnam), highest for green and raw Pu’erh teas (74.3 to 133.8 mg g^−1^, except for the DL-G-02 sample), and lowest for black teas (2.6 to 10.1 mg g^−1^). Overall, the values for Vietnamese green and oolong teas were higher than those reported in previous works [57,58]. For EGC concentrations, there was differentiation between oolong teas coming from two regions, Da Lat and Lam Ha (both in South Vietnam), with higher values observed for Da Lat’s oolong teas, against the values of EGCG. Higher concentration of EGC were found in green and raw Pu’erh teas (19.3 to 41.5), and the lowest was in black teas (6.1 to 8.7 mg g^−1^). The degree of difference in values of ECG, EC, and C was relatively low among tea types. However, total catechins (calculated from the sum of concentrations of five compounds C, EC, ECG, EGC, and EGCG) showed highest values for green and raw Pu’erh teas from Suoi Giang followed by green teas from Da Lat, oolong teas, and the lowest values were measured for black teas. Donlao et al. [18] reported three changing trends in flavonoids in green tea with change in drying temperature as follows: a decreasing trend recorded for EGC, EC, EGCG, and total catechins; an increasing trend observed for GC and GCG; and negligible change for C and CG. These observations were explained to be due to epimerisation of catechins into isomers, which increases the non-epistructure and decreases the epistructure compounds.

### 2.4. Pearson Correlation

The Pearson correlation was established to determine the relationships among the studied parameters such as TPCs, TFCs, DPPH, ABTS, FRAP, CUPRAC, and the contents of caffeine, gallic acid, and major catechins obtained from different teas in North and South Vietnam. Several good correlations were found with coefficients r^2^ > 0.800, but the corresponding scatter diagrams indicated not only linear correlations but also spurious correlations (Figure 1). In general, a better correlation between the TPC values and each antioxidant capacity was found when compared to TFCs. Corresponding to two groups of antioxidant capacities, for each group, TPCs were better correlated with ABTS than DPPH (0.976 vs. 0.940) and CUPRAC than FRAP (0.960 vs. 0.901). Following the same tendency, TFCs had a better correlation with ABTS than DPPH (0.896 vs. 0.790) and CUPRAC than FRAP (0.909 vs. 0.898). For ECG and EGCG, better correlations were found with DPPH than ABTS, and CUPRAC than FRAP. Moreover, ECG showed better association with antioxidant capacities than EGCG. Otherwise, among the four antioxidant mechanisms, this current study showed good correlations between CUPRAC and DPPH (0.954) and between CUPRAC and ABTS (0.967). In the study of Anesini et al. [10], a good correlation between TPCs and DPPH was presented for 12 commercial green and black teas in Argentina with r^2^ = 0.9144. Another study for 30 tea infusions from different types of tea in China [25] showed good correlations between TPCs and FRAP (r^2^ = 0.915) and between TPCs and ABTS (r^2^ = 0.946).

The antioxidant capacity of polyphenols is mainly due to their ability to act as hydrogen acceptors, reducing agents, and free radical scavengers [59,60]. Free radical scavengers are compounds capable of transferring hydrogen and electrons to free radicals [59]. Based on this definition, the antioxidant mechanisms can be divided into hydrogen atom transfer (HAT) ArOH + ROO∙ → ArO∙ + ROOH, and electron transfer (ET) ArOH → ArO^–^ + H^+^ then ArO^–^ + ROO∙ → ArO∙ + ROO^–^. The ET and HAT mechanisms always coexist and the balance between the two mechanisms depends on the structure of the antioxidant and pH [51]. DPPH scavenging, FRAP, and CUPRAC assays are three examples of antioxidant assays following the ET mechanism, while cationic radical scavenging assay using ABTS is reported to follow both ET and HAT mechanisms [51]. Consequently, ABTS would react to more compounds in tea and that could explain the better correlation between antioxidant capacity obtained from the ABTS free radical scavenging mechanism and TPCs than the other antioxidant assays.

### 2.5. Principal Component Analysis

Several studies have used principal component analysis (PCA) in the preliminary discrimination of teas with respect to their geographical origin and tea types [58,61,62]. Different parameters, including multi-elements, polyphenol composition, metabolites, antioxidant capacity, and stable isotopes, were analysed, following which chemometric methods were applied including PCA [63]. In this study, different variables consisting of two chemical parameters (TPCs and TFCs), four biological ones (DPPH, ABTS, FRAP, and CUPRAC), and contents of caffeine, gallic acid, and major catechins were set with 28 tea samples as observations to perform PCA. Different numbers of variables in the 28 tea samples were chosen as inputs into SIMCA-P 11 software.

Based on the two-dimensional (2D) graph of PCA score obtained with 6 variables for 11 tea samples presented in Table 1, the percentage variances of the two principal components are 92.2% (with eigenvalue 5.53) and 4.3%; thus, the cumulative variance is 96.5% (Figure 2a). The corresponding PCA score plot showed that tea samples could be divided into four groups composed of: (i) five green and raw Pu’erh teas from wild/ancient tea region in Suoi Giang commune (North Vietnam); (ii) two black teas from Suoi Giang; (iii) three oolong teas from Da Lat city and Lam Ha district (South Vietnam); and (iv) one green tea from Da Lat city. There was a clear differentiation in the PCA score plot between non-/low-oxidised teas (green and raw Pu’erh teas) and partly/fully oxidised teas (oolong and black teas). Additionally, there was a rather clear differentiation between green teas from Da Lat and green teas from Suoi Giang, but oolong teas from the two regions of Lam Dong Province showed no significant differentiation. Combining the values of caffeine, gallic acid, five major catechins, and total catechins (Table 2), a second 2D graphic of PCA score was obtained (Figure 2b) with percentage variances of PC1 and PC2 of 63.0% (with eigenvalue 6.93) and 19.2%, respectively; thus, the cumulative variance was 82.3%. The differentiation among tea groups was still clearly seen. However, the impact of new variables input in the PCA model changed the position of the groups (ii), (iii), and (iv) in the two left quarters of the ellipse (Hotelling’s T^2^ 95%). This was identified by the presence of EGC concentration in this PCA model. Additionally, the three compounds GA, C, and EC had weak impacts on the variation. Therefore, the removal of these parameters enabled the model to have better discrimination (74.2%, 19.6%, and 93.8% for PC1, PC2, and cumulative variance, respectively).

For the 28 tea samples presented in Table 2, the PCA score obtained with 5 variables consisting of CFI, EGC, EGC, EGCG, and total catechins is shown in Figure 3a. The percentage variances of the two first principal components are 53.1% (with eigenvalue 5.53) and 32.2%; thus, the cumulative variance is 85.3%. A similar study was performed by Yi et al. [58] in which 74 tea samples representing the 7 types of tea could be distinguished by using PCA, with 86.62% of the cumulative percent variance for the 3 first components. In the current study, the Biplot graph (Figure 3b) that overlays the score plot and the loading plot showed the impact of each variable on the two first components for the tea samples in which ECG, EGCG, and total catechins had high positive associations with PC1, which in turn helped to distinguish non-/low-oxidised teas (green and raw Pu’erh teas) and partly/fully oxidised ones (oolong and black teas). Meanwhile, for oolong and black teas, the EGC compound had a strong impact on the oxidation level.

These results prove that differences in latitude (North and South Vietnam), climate, types of tea (green, raw Pu’erh, oolong, and black), age (3–10, 100, and >200 years old), and processing (especially oxidation level) created significant differentiations in the tea composition and these variations can be identified through multivariate analysis with PCA.

## 3. Materials and Methods

### 3.1. Sample Collection, Storage, and Pre-Treatment

Three tea varieties have been investigated in this study: (i) called “Shan tea” for wild/ancient tea trees (>200 years old) grown in the mountainous area (Suoi Giang commune, Yen Bai Province, North Vietnam) from which green (SG-G), raw Pu’erh (SG-P), and black or red teas (SG-B) were produced; (ii) midland tea variety with smaller leaves than Shan tea cultivated in Da Lat city (Central Highland of Lam Dong Province, South Vietnam) for around 10 to 100 years to produce green teas (named DL-G samples); and (iii) Taiwanese oolong tea variety aged 3–10 years from Lam Ha district (LH) and Da Lat city (DL), situated both in Lam Dong Province (South Vietnam) with similar processing for these two regions. A total number of 28 dried tea products were collected in 2020 and 2022 (Appendix A) from green (G), raw Pu’erh (P), oolong (O), and black (B) teas. The green teas in the present study were referred to as non-oxidized due to the “killing green” step at high temperature for a few min to totally deactivate the oxidative enzymes inside tea leaves, rolling, and then drying in rotating drum dryer. In contrast, the raw Pu’erh teas were partly oxidized, rolled, and then dried under sunshine. The raw Pu’erh teas are used as raw materials to produce ripe or aged Pu’erh through specific fermentation processes. The oolong teas were around 20–30% oxidized and rolled into tight balls by drying several times.

Prior to sample extraction, the tea products were subjected to a pre-treatment procedure of grinding for homogenisation purposes. The homogenised tea samples were filled into non-permeable bags, sealed under vacuum, and stored under ambient conditions of the laboratory while avoiding exposure to direct sunlight until further experiments.

### 3.2. Sample Extraction

The tea extracts were obtained following ISO 14502-1 (2005) procedure with some modifications. Briefly, 0.2 (±0.001) g of the tea sample was weighed into a 15 mL polypropylene (PP) centrifuge tube. An amount of 10 mL of the extraction solvent consisting of 70% *v/v* methanol (MeOH, analytical grade, Merck, Darmstadt, Germany) in deionised water (DIW, Millipore, Billerica, MA, USA) was added and the mixture was vortexed for 1 min before heating in a water bath at 70 °C for 20 min. The sample was then centrifuged at 5000 rpm for 5 min, and the supernatant was transferred into a 50 mL volumetric flask. The extraction procedure was repeated one more time and the liquid was collected and added to a volumetric flask and made up to 50 mL with the extraction solvent (70% *v/v* MeOH in DIW). The liquid sample was then filtered through a 0.45 µm PTFE membrane and appropriately diluted with DIW for colorimetric assays and measurements to determine the total phenolic contents (TPCs), total flavonoid contents (TFCs), and Trolox equivalent antioxidant capacities (TEACs). Analytical results were expressed on a dried mass (DM) basis.

### 3.3. Method Analysis and Validation

#### 3.3.1. Total Phenolic Contents (TPCs)

Colorimetry was performed using the chemical reaction between the phenolic compounds and the Folin–Ciocalteu reagent (Merck, Darmstadt, Germany) as described in ISO 14502-1 (2005) with some minor changes. In brief, 0.50 mL of the diluted tea extract was pipetted in a 15 mL PP centrifuge tube to which 2.50 mL of 10% (*v*/*v*) Folin–Ciocalteu reagent was added and the mixture was gently shaken. The reaction mixture was left standing for around 3 to 8 min before adding 2.00 mL of 7.5% *w/v* Na_2_CO_3_ solution. The mixture was shaken gently and allowed to react for ~ 60 min. The absorbance was measured at 765 nm on a Shimadzu UV-1800 UV/Visible Scanning Spectrophotometer (Kyoto, Japan). The TPCs in tea were expressed as mg gallic acid equivalents/gram of the dried mass sample (mg GAE g^–1^ DM).

#### 3.3.2. Total Flavonoid Content (TFCs)

The total flavonoid contents (TFCs) of the diluted tea extracts were measured based on the aluminium chloride colorimetric procedure reported in our previous study [64] as briefly described below: 1.00 mL of the extract was added to a 15 mL PP centrifuge tube followed by adding 4.00 mL of DIW and 300 µL of 5% NaNO_2_. The mixture was shaken gently and left standing for 5 min before adding 300 µL of 5% AlCl_3_ solution and 2.00 mL of a 1 mol L^–1^ NaOH solution. Immediately, the reaction mixture was filled to 10.00 mL by DIW. Quercetin was used as the standard and the absorbance was measured at 510 nm on Shimadzu UV-1800 UV/Visible Scanning Spectrophotometer (Japan). The TFCs were reported as milligrams of quercetin equivalents per gram dried mass sample (mg QE g^–1^ DM).

#### 3.3.3. DPPH Assay

The DPPH assay has been commonly used to evaluate the TEACs in various food and plant matrices. In this study, the DPPH assay was carried out according to the method presented by Mishra et al. [65]. Briefly, a 0.609 mmol L^–1^ stock DPPH solution was prepared from 0.0120 (±0.0002) g of commercial DPPH (Merck, Germany) dissolved in 50 mL of MeOH. After being allowed to stand at the ambient temperature for 12–16 h for the stabilisation of radicals, the stock reagent solution was further diluted to attain an absorbance of around 0.98 ± 0.02 at 517 nm (quantification wavelength). An amount of 100 μL of the diluted tea extract (DIW for blank sample) was mixed with 3900 μL of the DPPH working solution. The mixture was left to stand for ~30 min in the dark before recording the absorbance at 517 nm. The % inhibition of DPPH free radicals was then calculated as [(absorbance of control − absorbance of sample)/(absorbance of control) × 100%]. Eventually, DPPH antioxidant activity was presented as the equivalent amount of Trolox in mg per gram dried mass sample (mg TE g^–1^ DM) based on the Trolox standard curve.

#### 3.3.4. ABTS Assay

The TEACs were determined in terms of ABTS radical scavenging activity using the procedure described by Marc et al. [66]. ABTS radical cations were produced by the reaction between a 7 mmol L^–1^ ABTS solution with a 2.45 mmol L^–1^ solution of potassium persulfate (K_2_S_2_O_8_). The mixture was allowed to stand in the dark at the ambient temperature for 12–16 h for the stabilisation of radicals prior to use. The ABTS radical cation solution (stable for 2 days) was used to prepare the working ABTS reagent solution by further diluting with 0.3 mol L^–1^ acetate buffer (pH 4.5) to achieve an absorbance of 0.98 ± 0.03 at 734 nm (quantification wavelength). An amount of 200 μL of the diluted tea extract (DIW for blank sample) was added to 4000 μL of the working ABTS solution. The mixture was then gently shaken and protected from direct light for 7 min before measuring the absorbance at 734 nm. The % inhibition of ABTS free radicals was then calculated as [(absorbance of control − absorbance of sample)/(absorbance of control) × 100%]. Eventually, ABTS antioxidant activity was presented as the equivalent amount of Trolox in mg per gram dried mass sample (mg TE g^–1^ DM) based on the Trolox standard curve.

#### 3.3.5. FRAP Assay

For ferric reducing antioxidant power (FRAP) experiments, the procedure used was based on the study by Pellegrini et al. [67]. The working FRAP reagent was prepared by mixing 300 mmol L^–1^ acetate buffer (pH 3.6), 10 mmol L^–1^ solution of 2,4,6-Tris(2-pyridyl)-s-triazine (TPTZ), and 20 mmol L^–1^ ferric chloride (Merck, Germany) in the volume ratio of 10:1:1. An amount of 100 μL of the diluted tea extract sample (DIW for blank sample) was added to 3600 μL of the working FRAP solution. The mixture was shaken gently and left standing for 15 min before measuring the absorbance at 593 nm. Note that the 10 mmol L^–1^ TPTZ solution was prepared by dissolving 0.1562 (±0.0001) g of TPTZ (Merck, Germany) in 100 mL of DIW containing 0.17 mL of concentrated hydrochloric acid HCl (Merck, Germany). The FRAP activity was presented as equivalent amount of Trolox in mg per gram dried mass sample (mg TE g^–1^ DM) based on the Trolox standard curve.

#### 3.3.6. CUPRAC Assay

In the cupric ion reducing antioxidant capacity (CUPRAC) experiments, the reduction power of tea samples was determined using the procedure described by Apak et al. [68]. In short, 1.00 mL of 10 mmol L^–1^ cupric chloride (Merck, Germany), 1.00 mL of 7.5 mmol L^–1^ neocuproine in ethanol (Merck, Germany), and 1.00 mL of ammonium acetate buffer (38.54 g of ammonium acetate dissolved in 500 mL of DIW, pH 7.0) were added to a glass reaction tube. Next, 0.25 mL of the diluted tea extract (DIW for blank sample) and 0.875 mL of DIW were added. The mixture was shaken gently and left standing for 1 h at room temperature prior to absorbance measurement at 450 nm. The CUPRAC activity was presented as equivalent amount of Trolox in mg per gram dried mass sample (mg TE g^–1^ DM) based on the Trolox standard curve.

#### 3.3.7. Quantification of Caffeine, Gallic Acids, and Major Catechins by UPLC-MS/M

A Xevo TQD UPLC-MS/MS system (Waters, Milford, MA, USA) and an ACQUITY UPLC BEH C18 Column 130 Å, 1.7 µm, 2.1 mm × 100 mm (Waters, Milford, MA, USA) were used to identify and quantify caffeine (CFI), gallic acid (GA), and the five major catechins (–)-epigallocatechin (EGC), (–)-epicatechin (EC), (+)-catechin (C), (–)-epigallocatechin gallate (EGCG), and (–)-epicatechin gallate (ECG) in multiple reaction monitoring (MRM) mode by mass spectrometry (Appendix A). All standards were purchased from Extrasynthese (Genay, France). The stock solutions were prepared in MeOH at the concentration of 1 mg mL^−1^ and stored at −18 °C. The working solutions were diluted in MeOH/H_2_O (1/1, *v*/*v*) and prepared fresh each day. The mobile phases of deionised water (A-DIW) and acetonitrile (B-MeCN, HPLC grade, Merck, Germany) were both supplemented with 0.1% formic acid. The injection was performed with 0.3 µL volume at 0.10 mL min^–1^ flow rate and 40 °C column temperature. The following gradient was employed: 0.0–0.5 min at 95% A; 0.5–5.0 min from 95% to 5% A and kept for 0.5 min; 5.5–6.0 min from 5% to 95% A; 6.0–9.0 min kept at 95% A. For this analysis, the sample solutions were extracted as described in ISO 14502-1 (2005) with some minor changes. 

#### 3.3.8. Method Validation

All the analytical methods mentioned above were validated according to the guidelines in Appendix F: Guidelines for Standard Method Performance Requirements (AOAC Official Methods of Analysis, 2016) as follows: (i) estimating the limit of detection (LOD) and limit of quantification (LOQ); (ii) construction of the calibration curve and working range; (iii) evaluation of repeatability and reproducibility; and (iv) evaluation of recovery. The results are presented in Appendix A.

### 3.4. Statistical Data Analysis

All statistical techniques including calculation of Pearson correlation were performed at 5% significance level by using the R version 4.1.2 [69].

Principal component analysis (PCA) was applied to explore the relation among the variables for different types of tea. The PCA was performed by SIMCA-P 11 software (Umetrics, Umeå, Sweden).

## 4. Conclusions

For teas produced from the same tea variety, tree age (wild/ancient), and growing conditions in Suoi Giang commune (Yen Bai Province, North Vietnam), TPCs and TFCs of green and raw Pu’erh teas showed higher values than black teas due to the oxidation step in tea processing. The difference in TPCs between oolong teas from Lam Dong Province (South Vietnam) was less important than the difference in TFCs, probably caused by the longer drying time in oolong tea processing and also the different tea varieties. Green teas from Suoi Giang (>200 years old and growing in the wild) showed higher values of TPCs and TFCs than green teas from Da Lat (around 100 years old and grown using conventional agriculture). In terms of antioxidant capacities according to the four mechanisms, DPPH, ABTS, FRAP, and CUPRAC, the results obtained for green and raw Pu’erh teas from the wild/ancient tea region of Suoi Giang (North Vietnam) showed minor differences between these two types of tea. The concentrations of caffeine varied from 13.6 to 54.4 mg g^−1^ and did not depend on the type of tea. Total catechins (calculated from the sum of concentrations of the five compounds, C, EC, ECG, EGC, and EGCG) showed the highest values for green and raw Pu’erh teas from Suoi Giang followed by green teas from Da Lat, oolong teas, and the lowest values were obtained for black teas. Several good Pearson’s correlations were found (r^2^ > 0.9) for positive associations between TPCs, TFCs, DPPH, ABTS, FRAP, and CUPRAC, and major catechins such as ECG and EGCG. There was a clear differentiation in the PCA score plot between non-/low-oxidised teas (green and raw Pu’erh teas) and partly/fully oxidised ones (oolong and black teas). These results prove that differences in latitude (North and South Vietnam), climate, types of tea (green, raw Pu’erh, oolong, and black), age (3–10, 100, and >200 years old), and processing (especially oxidation level) can create significant differentiations in tea composition using which other methods can be developed to distinguish tea quality and authenticity.

## Figures and Tables

**Figure 1 molecules-28-03470-f001:**
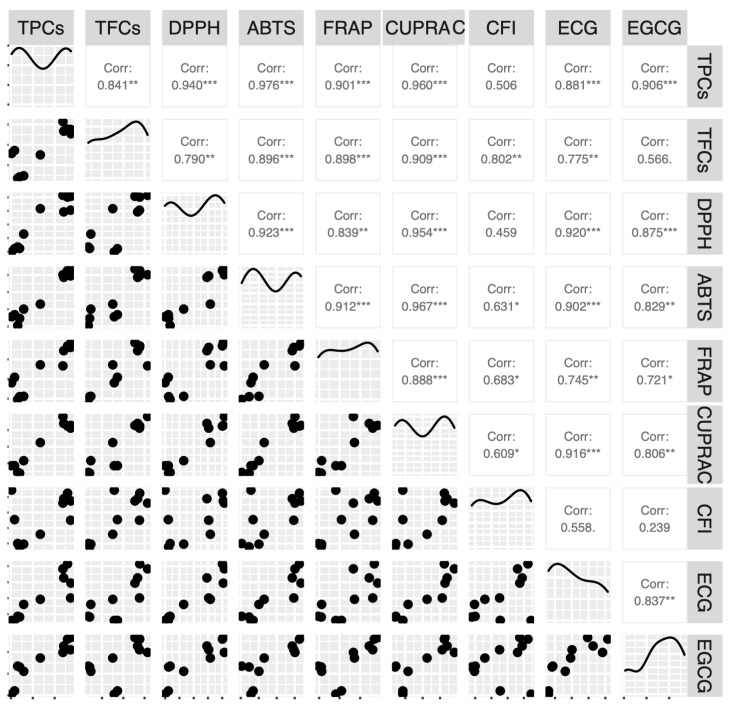
Pearson correlation coefficients among the determined parameters for dried tea products of different geographical origins and varieties in Vietnam. (Note: ***, **, * represent significance levels of 0.1%, 1%, 5%, respectively).

**Figure 2 molecules-28-03470-f002:**
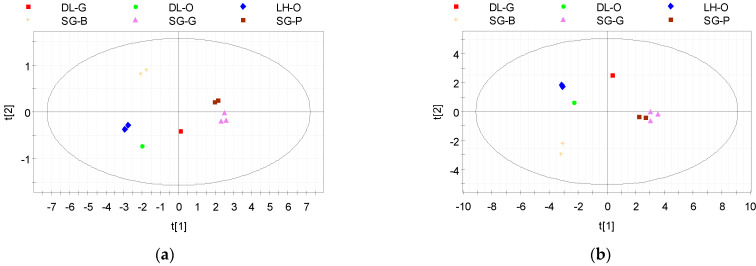
The PCA scores of the first two principal components for 11 tea samples in Vietnam based on (**a**) 6 variables: TPCs, TFCs, DPPH, ABTS, FRAP, and CUPRAC with 96.5% of the cumulative variance of the two first components; and (**b**) 14 variables: TPCs, TFCs, DPPH, ABTS, FRAP, CUPRAC, caffeine, gallic acid, 5 major catechins, and total catechins with 82.3% of the cumulative variance of the two first components.

**Figure 3 molecules-28-03470-f003:**
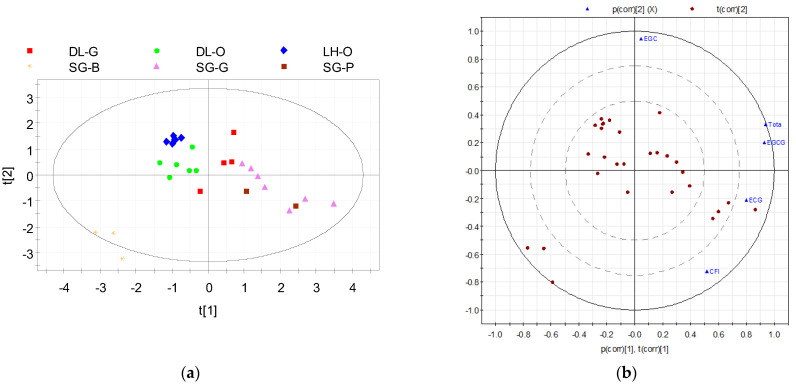
The PCA scores (**a**) and Biplot graph (**b**) of the first two principal components for 28 tea samples in Vietnam based on 5 variables (CFI, EGC, EGC, EGCG, and total catechins) with 85.3% of the cumulative variance of two first components.

**Table 1 molecules-28-03470-t001:** The variation of TPCs, TFCs, and antioxidant activities (DPPH, ABTS, FRAP, CUPRAC) in dried tea products of different geographical origin and the different tea varieties in Vietnam.

Tea Sample	TPCs	TFCs	DPPH	ABTS	FRAP	CUPRAC
	(mg GAE g^–1^ DM)	(mg QE g^–1^ DM)	(µmol TE g^–1^ DM)	(µmol TE g^–1^ DM)	(µmol TE g^–1^ DM)	(µmol TE g^–1^ DM)
DL-G-01	217.2 ± 1.1 a	125.3 ± 1.3 a	2541.3 ± 31.6 a	3651.2 ± 37.9 a	2361.1 ± 23.9 a	3625.5 ± 31.9 a
DL-O-01	179.2 ± 1.7 b	73.7 ± 0.6 b	2074.0 ± 35.8 b	3505.6 ± 39.2 b	1561.1 ± 26.7 b	3079.2 ± 33.4 b
LH-O-02	165.4 ± 3.6 c	67.7 ± 0.7 c	1835.6 ± 31.3 c	3025.7 ± 26.5 c	1481.2 ± 22.6 c	2726.2 ± 23.5 c
LH-O-06	169.8 ± 0.5 c	70.9 ± 1.3 d	1809.8 ± 30.0 c	3233.2 ± 34.5 d	1552.0 ± 23.4 b	2704.9 ± 22.3 c
SG-G-01	287.2 ± 5.9 d	177.3 ± 0.7 e	2769.1 ± 27.5 d	4675.1 ± 33.4 e	2927.6 ± 23.5 d	4141.2 ± 33.8 d
SG-G-04	279.5 ± 0.6 d	188.6 ± 0.2 f	2754.0 ± 20.4 d	4630.3 ± 34.8 e	2844.5 ± 25.0 e	4121.1 ± 38.1 d
SG-G-07	268.9 ± 1.5 e	207.1 ± 0.3 g	2787.4 ± 31.9 d	4506.2 ± 35.9 f	2334.4 ± 25.4 a	4410.6 ± 37.4 e
SG-P-01	270.8 ± 7.6 e	183.6 ± 1.4 h	2482.0 ± 26.3 a	4431.7 ± 35.1 f	2721.4 ± 28.9 f	4201.1 ± 33.7 d
SG-P-02	286.5 ± 6.2 d	188.0 ± 0.7 f	2513.5 ± 31.3 a	4474.3 ± 36.9 f	2806.3 ± 25.3 e	4060.7 ± 32.2 d
SG-B-01	151.4 ± 1.5 f	128.8 ± 0.1 i	1745.4 ± 26.7 c	3283.6 ± 35.5 d	1906.6 ± 23.6 g	2927.3 ± 39.9 f
SG-B-02	158.2 ± 3.1 f	136.1 ± 1.1 j	1803.8 ± 25.0 c	3343.4 ± 38.4 d	2043.4 ± 30.1 h	2928.8 ± 32.6 f
*p*-value	6.19 × 10^−25^	3.51 × 10^−38^	7.98 × 10^−25^	6.89 × 10^−27^	1.56 × 10^−28^	7.67 × 10^−28^
Green tea [28]	63.87–80.27	20.90–35.17				
Black tea [28]	56.63–76.93	19.07–33.70				
Green orthodox [29]	271					
Green CTC [29]	250.7–268.5					
Black orthodox [29]	222.5					
Black CTC [29]	174.5–206.5					
Oolong orthodox [29]	261.5					
Green tea [30]	113.7–141.2					
Black tea [30]	60.6–84.9					
Oolong tea [30]	75.0–90.9					

Notes: Reference data were compiled for teas originating from Sabah (Malaysia) [28], Kenya [29], and Boh (Malaysia), Sea Dyke (China), An Xi (China) [30]. Data in the same column with different letters indicate the significant differences at significance level of 5% (*p*-value < 0.05) based on Tukey HSD test.

**Table 2 molecules-28-03470-t002:** Concentration (mg g^−1^ DM ± standard deviation) of caffeine (CFI), gallic acid (GA), catechin (C), epicatechin (EC), epicatechin gallate (ECG), epigallocatechin (EGC), and epigallocatechin gallate (EGCG), and total catechins (TCs) in various dried tea products of different geographical origins and varieties in Vietnam.

Tea Sample	CFI	GA	C	EC	ECG	EGC	EGCG	TCs
DL-G-01	25.94 ± 0.02	4.07 ± 0.01	5.21 ± 0.04	12.77 ± <0.01	45.97 ± 0.02	19.33 ± 0.07	86.31 ± 0.03	169.6
DL-G-02	23.50 ± 0.07	5.49 ± 0.40	7.18 ± 0.26	12.50 ± 0.16	22.63 ± 0.15	33.56 ± 0.13	39.01 ± 0.03	114.9
DL-G-03	27.56 ± 0.01	7.84 ± 0.03	7.57 ± 0.02	14.22 ± 0.03	33.42 ± 0.06	22.24 ± 0.06	74.30 ± 0.01	151.7
DL-G-04	21.45 ± 0.10	3.09 ± 0.06	2.03 ± 0.75	2.27 ± 0.01	30.94 ± 0.13	29.01 ± 0.42	82.37 ± <0.01	146.6
DL-O-01	19.56 ± 0.15	1.24 ± <0.01	2.12 ± 0.05	4.54 ± 0.47	30.14 ± 0.22	16.15 ± 0.15	55.50 ± 0.91	108.4
DL-O-02	17.34 ± 0.02	1.60 ± 0.02	1.42 ± 0.01	9.24 ± <0.01	28.83 ± 0.03	8.56 ± 0.01	55.13 ± 0.02	103.2
DL-O-03	13.64 ± 0.20	0.30 ± <0.01	4.06 ± 0.04	1.53 ± 0.02	26.26 ± 0.39	28.97 ± 0.41	50.66 ± 0.04	111.5
DL-O-04	17.89 ± 0.07	6.05 ± 0.36	2.13 ± 0.53	1.95 ± <0.01	25.88 ± 0.15	22.34 ± 0.13	41.44 ± 0.99	93.7
DL-O-06	20.17 ± 0.33	2.16 ± <0.01	3.14 ± 0.05	5.62 ± 0.05	40.15 ± 0.12	21.15 ± 0.12	54.42 ± 0.60	124.5
DL-O-08	17.97 ± 0.01	7.42 ± 0.36	6.24 ± 0.33	2.12 ± 0.02	30.08 ± 0.15	33.05 ± 0.11	42.36 ± 0.86	113.9
LH-O-01	15.99 ± 0.04	2.74 ± <0.01	1.26 ± 0.01	9.36 ± 0.01	39.57 ± 0.05	7.85 ± 0.05	65.43 ± 0.05	123.5
LH-O-02	19.95 ± 0.01	2.75 ± 0.03	1.76 ± 0.01	8.93 ± 0.04	41.15 ± 0.03	8.70 ± 0.03	67.50 ± 0.04	128.0
LH-O-03	19.17 ± 0.01	1.56 ± 0.02	1.64 ± <0.01	8.32 ± 0.04	38.33 ± 0.03	8.10 ± 0.03	62.87 ± 0.04	119.3
LH-O-05	17.15 ± 0.07	2.61 ± 0.03	1.47 ± 0.01	8.39 ± 0.02	38.55 ± 0.09	7.62 ± 0.02	59.38 ± 0.03	115.4
LH-O-06	17.86 ± 0.02	2.50 ± 0.02	1.38 ± <0.01	9.16 ± 0.01	38.99 ± 0.01	8.25 ± 0.01	65.55 ± 0.05	123.3
LH-O-07	18.38 ± 0.03	2.12 ± 0.02	2.56 ± <0.01	8.70 ± 0.03	39.26 ± 0.04	8.17 ± 0.02	63.82 ± 0.04	122.5
SG-G-01	47.47 ± 0.29	1.88 ± 0.03	2.82 ± 0.05	5.55 ± 0.06	21.25 ± 0.20	29.45 ± 0.44	133.80 ± 1.60	192.9
SG-G-02	33.25 ± 0.13	2.21 ± 0.02	2.32 ± 0.02	9.10 ± 0.12	23.41 ± 0.13	32.14 ± 0.15	99.15 ± 0.14	166.1
SG-G-03	22.98 ± 0.13	10.25 ± 0.03	2.52 ± 0.29	7.32 ± 0.37	22.19 ± 0.12	20.13 ± 0.15	120.39 ± 0.47	172.6
SG-G-04	52.49 ± 0.02	2.01 ± 0.02	3.25 ± 0.05	8.89 ± 0.01	23.24 ± 0.01	41.46 ± 0.43	130.16 ± 1.70	207.0
SG-G-05	35.44 ± 0.25	0.55 ± 0.05	3.15 ± 0.02	14.42 ± 0.04	30.12 ± 0.12	29.11 ± 0.45	90.41 ± 0.25	167.2
SG-G-06	21.43 ± 0.08	9.07 ± 0.09	4.67 ± 0.46	9.94 ± 0.36	25.93 ± 0.49	23.25 ± 0.47	102.44 ± 0.52	166.2
SG-G-07	46.09 ± 0.37	1.24 ± 0.01	2.14 ± <0.01	12.87 ± 0.11	19.55 ± 0.07	38.34 ± 0.39	99.24 ± 0.17	172.1
SG-P-01	48.93 ± 0.72	2.59 ± 0.01	4.24 ± <0.01	7.76 ± 0.01	21.50 ± 0.23	32.69 ± 0.16	114.17 ± 1.59	180.4
SG-P-02	35.02 ± 1.14	2.51 ± 0.05	4.66 ± 0.06	6.90 ± 0.24	19.92 ± 0.41	20.15 ± 0.32	105.46 ± 1.45	157.1
SG-B-01	54.44 ± 0.65	3.50 ± 0.02	1.74 ± 0.01	2.93 ± <0.01	10.20 ± 0.03	6.11 ± 0.02	2.60 ± 0.01	23.6
SG-B-02	35.41 ± 1.24	3.54 ± 0.12	1.82 ± 0.01	3.94 ± 0.06	10.41 ± 0.14	6.14 ± 0.14	10.12 ± 0.12	32.5
SG-B-03	22.44 ± 0.06	6.15 ± 0.61	1.73 ± 0.17	4.30 ± 0.02	4.17 ± 0.02	8.70 ± 0.33	3.30 ± 0.01	22.2

## Data Availability

All data are available in the manuscript.

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
