# Peer review of "Characteristics and Relationships between Total Polyphenol and Flavonoid Contents, Antioxidant Capacities, and the Content of Caffeine, Gallic Acid, and Major Catechins in Wild/Ancient and Cultivated Teas in Vietnam"

_molecules, 2023, doi:10.3390/molecules28083470_

Round 1

Reviewer 1 Report

Dear Authors,

The article submitted for review entitled  “Characteristics and relationships of total polyphenol and flavonoid contents, antioxidant capacities, caffeine, gallic acid and major catechins in wild/ancient and cultivated teas in Vietnam” is a very interesting comparison of tea with various degrees of processing obtained from various plants located in several regions of Vietnam. The authors analysed 28 different types of tea and characterized in great detail the level of various phenolic compounds and the antioxidant potential of leaves, they also determined the correlation relationships between the measured parameters. This study has a high cognitive value and will contribute to the enrichment of knowledge on the factors influencing the chemical composition of tea and its health-promoting effects, but before publishing, the authors should clarify some elements of the paper and supplement others.

I have indicated detailed comments in the attached text of the revised manuscript, and below I indicate the most important of them:

1. The authors write in the abstract about the study of 28 tea samples, the results of which they summarized in unpublished materials, but in the article they characterize only selected of them. It should be indicated what motivated this choice. It is also worth pointing out whether, in order to better compare the properties of tea, it was not possible to sample the same types of tea (e.g. green, pu-erh, oolong and black) from different regions of Vietnam? Did individual provinces specialize only in the production of certain (not all) types of tea due to the degree of its processing?

2. The paper is sometimes difficult to read, which results from the use of a large number of abbreviations (defining the place of origin and type of tea, as well as the names of the tested distinguishing features), which, in combination with giving the values of the marked parameters and their units, makes it difficult to understand the meaning of the sentence and authors' mindset. Examples of improving  in this area are given in the notes to the manuscript.

3. Despite the advanced statistical analysis in the form of specific correlation relationships, and especially the PCA analysis, which perfectly illustrates the found relationships, the paper lacks an analysis of the significance of differences between the determined parameters (Tables 1 and 2). Doing it would make it easier for the authors to describe the obtained dependencies. The very small dispersion of the obtained results (low values of standard deviations) is also puzzling, which rarely happens in the case of the determination of low-stable polyphenolic compounds and antioxidant activity. How did the authors manage to do it? Were the values of standard deviations measured for parallel samples taken several times from the leaves of a given tea? Or were the leaves collected once, and the given values refer to the repeatability of the extraction process or measurements themselves?

4. The conclusions in the current version of the article are more a summary of the results, and not typical conclusions. General conclusions from the conducted research should be provided in this chapter. This should be a specific answer to the objectives of the study, on the basis of which other authors planning investigations in this area will be able to supplement their knowledge and clarify and update their research assumptions.

More specific notes:

1. Is pu-erh tea oxidized? The authors provide various information on this subject.

2. The order of data presented in Fig. 1 and in Tab. 1 should be the same.

3. Table 2 is illegible, so I was unable to analyze the dependencies discussed by the authors in lines 286-308. To improve its readability I suggest entering the values in the table with an accuracy of 2 decimal places. It is also a good idea to leave spaces between the lines with different parameters. Or maybe it is worth to consider removing standard deviations and adding letters showing the significance of differences between individual values?

Author Response

Dear Reviewer,

I thank you very much for your comments. Please find my point-to-point response in the attached file and also in the revised manuscript.

Best regards,

Dr. Nhu-Trang

Reviewer 2 Report

This research investigates tea polyphenol contents, phenolic profiles, antioxidant capacities of 28 different tea samples collected in Vietnam. This work is substantial and interesting, and maybe assists in utilizing Vietnam teas as source of natural polyphenols. However, the presentation is long and not good, revisions are needed, questions and suggestions are as following.

1.        Title and key words could be reduced for accuracy and exactness.

2.        Can you please clearly state the research significance or contribution for this field in the abstract or the last paragraph of the introduction?

3.        Tea originated from China, so much deep research on tea polyphenols has been carried out. Did you compare the results of the studied teas in Vietnam with those cultivated in China? Can you please clearly clarify the novelty of the research in abstract or other sections?

4.        For the introduction, it’s tedious and too general, the authors did not summary the right focus of the research, please reconsider. In the same, discussion should be improved.

5.        How were the teas processed? processing method is very important for tea quality and tea polyphenols. in lab or commercial factories? Please clarify in the method section or in supplementary file.

6.        How was the antioxidant capacity calculated for all the four assays?

7.        All standards were purchased from Extrasynthese (France). Line 193-197, the material and standards should be described in material and reagents.

8.        What are the MS conditions in the quantification and identification the target compositions?

9.        How many phenolic standards did you used in this study? Only procyanidin B2 standard was mentioned in line 87.

10.    I am confused by Table 1 and Figure 1, the data are the same. Why used table together with figure?

11.    Figure 2, the picture is obscure.

12.    Line 149-151, are you sure about your conclusion? what are the probable reasons for the higher TPCs of Vietnamese tea than those from other countries?

13.    grammatical mistakes should be checked in the whole manuscript.

14.    Some unessential discussions could be deleted, for example, line 24-30 in section 4.2. Please check for avoiding too general descriptions.

15.    Please give a concise conclusion for your research in section 5.  

Author Response

(The authors gave the same response as above.)
